# Heavy Metal Accumulation in Sediments of Small Retention Reservoirs—Ecological Risk and the Impact of Humic Substances Distribution

Lilianna Bartoszek *, Renata Gruca-Rokosz [ID], Agnieszka Pękala [ID] and Joanna Czarnota [ID]

Department of Environmental Engineering and Chemistry, Faculty of Civil and Environmental Engineering and Architecture, Rzeszów University of Technology, al. Powstańców Warszawy 12, 35-959 Rzeszów, Poland
* Correspondence: bartom@prz.edu.pl

**Abstract:** Anthropogenic pollutants that accumulate in bottom sediments may pose a serious threat to the aquatic environment and humans. The aim of the study was to determine the ecological risk related to the contamination of sediment with heavy metals and the relationship between the accumulation of heavy metals and various granulometric fractions and humic substances in the bottom sediments of small retention reservoirs located in catchments of varying anthropopressure. The research objects were five small dam reservoirs located in south-eastern Poland. The sediments of the reservoir exposed to the greatest anthropopressure from the catchment area posed a serious threat to aquatic organisms feeding at the bottom. The bottom sediments of the remaining reservoirs showed a low level of potential toxicity (or non-toxicity). The observed relationship between the enrichment of sediments with organic matter (OM) and the increased risk of their ecotoxic impact on aquatic organisms was determined by excessive exposure to heavy metal contamination. The sand content did not appear to have a clear effect on the metal accumulation, although it was associated with enrichment in OM. Due to diverse environmental conditions, it was not possible to unequivocally confirm that the accumulation of heavy metals in the sediments of small retention reservoirs directly depends on the content of organic matter and humic substances, but such relationships were observed in most of the objects.

**Keywords:** small dam reservoirs; bottom sediments; heavy metals; ecological risk analysis; organic matter; humic substances (HS) fractionation; HS-metals connections



## 1. Introduction

The progressive degradation process of water reservoirs has become one of the most important problems regarding water ecosystems, which has a huge impact on the quality of Polish and global resources and the possibility of their recreational use. Depositing a significant part of substances in bottom sediments, including many pollutants dangerous for the biocenosis introduced by inflows and surface runoff, is a natural defense mechanism against the progress of the degradation of water reservoirs. Thus, bottom sediments have become an important element in determining the severity of aquatic ecosystem degradation [1–4].

The matter that accumulates in sediments may come directly from external sources, as well as arise as a result of biological and chemical processes that take place within the reservoir [5]. The number of substances accumulated in the bottom sediments of dam reservoirs depends on many factors, such as the intensity of water flow, the depth of the facility, the physical and chemical conditions in the water and in the water-sediment interface, the granulometric and chemical composition of the sediments, but most of all the anthropopressure of the catchment area [6–8].

Among the matter accumulated in the sediments, the most important is their contamination with toxic substances, i.e., heavy metals and organic compounds of anthropogenic

origin, belonging to the group of so-called non-trophic pollutants. Non-trophic pollutants are introduced into waters along with municipal and industrial sewage, surface runoff from urban and industrial areas, from agricultural land (plant protection products), and communication routes [1,9–12]. Dust from the combustion of fossil fuels and biomass can also be a significant source [13–15]. Due to their low solubility, they accumulate in bottom sediments, where they are associated with organic matter, coal dust, and soot [13]. Heavy metals, unlike organic pollutants, do not biodegrade; they only transform and remain in the environment in constant circulation [9,16]. Some of the metals in bottom sediment occur in the structure of minerals that are relatively resistant to changing environmental conditions (e.g., heavy minerals) and do not pose a threat to the biosphere. The remaining part is present in the form of carbonates, phosphates, sulphides, oxides, etc., and they are also adsorbed by clay minerals, organic matter, and hydrated iron and manganese oxides [1,17]. Bottom sediments have varying abilities to accumulate pollutants from water due to their sorption capacity and grain size [18]. By having toxic effects on aquatic organisms (including bacteria and algae), heavy metals can inhibit biochemical processes such as the photosynthesis and/or biodegradation of organic matter, which results in disrupting water self-purification processes. The ecotoxicity of sediments contaminated with heavy metals and selected organic compounds of anthropogenic origin for aquatic organisms is determined by the method of numerical indicators of sediment quality TEC (threshold effect concentration) and PEC (probable effect concentration) [17,19–21]. When the content of a given substance in the sediment exceeds the TEC threshold value, harmful effects on the biocenosis are already possible. Toxic effects of a substance on aquatic organisms occur continuously when the PEC value is exceeded.

In addition to pollutants of anthropogenic origin, natural organic matter (NOM) is accumulated in the bottom sediments of water reservoirs, which are plant and animal remains, humic substances, metabolic products of microorganisms, and organic compounds from a group of carbohydrates, amino acids, acids, esters, lipids, dyes, or lignins [22]. Naturally occurring organic substances in the environment are usually more easily degradable. Due to their high molecular weight and complicated chemical structure, humic substances decompose slowly [23–25]. The large number of various functional groups (mostly carboxylic and phenolic) contained in the HS molecules is responsible for their ability to bind metals in the form of salts and complex connections, as well as strong sorption properties against toxic organic pollutants [15,26–28]. The role of humic substances as carriers of heavy metals in the environment is known; however, it is often overlooked in studies of aquatic ecosystems. Usually, the relationship between metals and total organic matter is taken into account. Humic acids (HA) can especially form stable organo-mineral complexes thanks to their physicochemical properties and biochemical stability, contributing to the reduction of the bioavailability of these pollutants [29]. Humic substances may arise within the reservoir and be introduced along with the surface runoff in a dissolved form (fulvic and humic acids) and suspended (humins) from peat bogs, wooded areas, and soils enriched with humus [30]. They are common in the aquatic environment and may constitute 60–90% of dissolved organic carbon in surface waters and even approx. 90% of the total organic matter content in lake sediments [25,31,32]. In an anaerobic environment, both in dissolved and solid form, they can act as an electron acceptor in the process of decomposition of organic pollutants [33].

The increase in the frequency of extreme weather phenomena, such as prolonged periods of drought or torrential rains, as well as decreasing freshwater resources, force us to increase water retention activities, e.g., through small reservoirs. According to an estimated assessment, there are currently over 16 million small bodies of water in the world [34]. Many of them are unsuitable as a source of drinking water and for recreational purposes due to advanced degradation [7]. Therefore, there is a need to stop and counteract the progress of their degradation, both in the case of existing and newly built reservoirs, which generates a need for continuous research in this direction.

The aim of the study was to determine the ecological risk related to the contamination of sediment with heavy metals and the relationship between the accumulation of heavy metals and various granulometric fractions and humic substances in the bottom sediments of small retention reservoirs located in catchments of varying anthropopressure.

## 2. Materials and Methods

### 2.1. Study Area and Sampling Strategy

Sediment samples were collected from 11 sampling sites within 5 small artificial reservoirs in south-eastern Poland (Figure 1) four times in the period from May to September 2013 and five times in the same period of 2014 (9 samples for each site). Basic morphological parameters of the studied reservoirs and the features of their catchments are presented in Table 1. The sediments were collected from three research sites on the Kamionka reservoir (near the tributary, near the center of the reservoir, and near the dam) and two study sites on the remaining reservoirs (near the tributary, near the dam) (Figure 1). Three sediment samples were collected at each site. For laboratory tests, the top layer (0–5 cm) of the sediment from each sample was taken and averaged. Part of the sediments was dried at 60 °C (for 48 h), homogenized, and stored in tightly closed PE bags at 4 °C, protected from light. As far as possible, the sediments were analyzed on an ongoing basis. The remaining part of the sediments was used for particle size analysis.

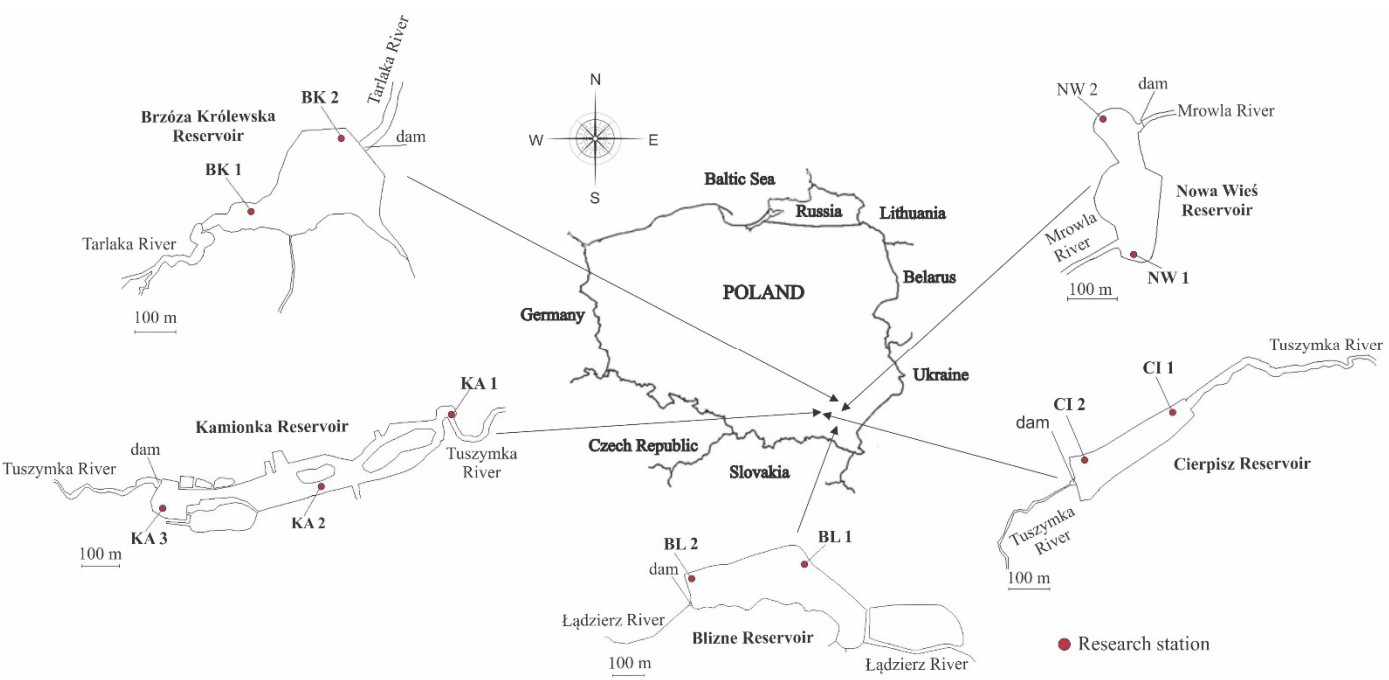

**Figure 1.** Location of reservoirs and sampling stations on the Cierpisz, Brzóza Królewska, Nowa Wieś, Kamionka, and Blizne reservoirs.

**Table 1.** Hydromorphological parameters and characteristics of the catchments of the analyzed small retention reservoirs.

| Parameters | Cierpisz | Brzóza Królewska | Nowa Wieś | Kamionka | Blizne |
|---|---|---|---|---|---|
| Year of construction | 1953 | 1978 | 1977 | 1957 | 2002 |
| Year of reclamation (desludging) | 1990–1991 | 1996 | - | 2007 | - |

**Table 1.** *Cont.*

| Parameters | Cierpisz | Brzóza Królewska | Nowa Wieś | Kamionka | Blizne |
|---|---|---|---|---|---|
| Coordinate | 50°09′ N 21°43′ E | 50°14′ N 22°19′ E | 50°06′ N 22°03′ E | 50°08′ N 21°40′ E | 49°44′ N 22°00′ E |
| Volume ($10^3$ $m^3$) | 22 | 50 | 75 | 105 | 137 |
| Mean depth (max.) (m) | 0.9 (1.5) | 0.7 (1.5) | 1.0 (3.0) | 1.5 (3.0) | 1.6 (3.9) |
| Area (ha) | 2.3 | 7.05 | 3.0 | 7.0 | 8.66 |
| Hydraulic retention time (days) | 1.2 | 2.5 | 1.3 | 4.8 | 18.0 |
| Catchment area ($km^2$) | 54.5 | 30.4 | 208.1 | 90 | 12.0 |
| Catchment type | Agricultural—forest | forest—agricultural with buildings | pasture—agricultural with buildings | pasture—agricultural—forest with buildings | pasture—forest—agricultural |
| Catchment shape | lowland | upland | upland | lowland | mountainous |

*2.2. Sediment Analysis*

The grain size analysis of bottom sediments was carried out using the sieve-areometric method. Areometric analysis was used to determine the boundary fraction content for particles with a diameter of 0.002 to 0.063 mm. Sieve analysis was used for fractions with particle diameters greater than 0.063 mm [35]. On the basis of the residue after ignition (at 550 °C for 4 h), the content of organic matter (OM) was calculated. The pH of the sediments ($pH_{KCl}$) was determined with the potentiometric method in a colloidal KCl suspension (1 mol·$dm^{-3}$). Mineralization of the sediments was carried out in concentrated $HNO_3$ in a high-pressure (pressure range 2–4.5 MPa) microwave mineralizer according to the procedure developed by the producer of the device (UniClever II, Plazmatronika, Poland). The content of heavy metals in mineralizates, i.e., chromium, zinc, cadmium, copper, nickel, and lead, was determined with the use of an ICP spectrometer (Integra, GBC). The certified reference material CRM 16-050 was used to check the quality of trace element analyses. The analysis results of the three quality control samples showed satisfactory compliance with the certified values. The average recovery percentage for individual metals was: 94.2% for Cd, 106.8% for Pb, 92.6% for Cr, 105.5% for Cu, 94.7% for Ni, and 109.6% for Zn.

The modified Griffith-Schnitzer fractionation method was used to determine the content of humic substance fractions in the bottom sediments [30,36,37]. An aliquot of 2–5 g of the sediment was shaken with 50 $cm^3$ of NaOH solution (0.1 mol·$dm^{-3}$) for 16 h in a previously weighed centrifuge tube. Before shaking, the free space of the tube was purged with $N_2$. The extract was separated from the sediment. In part of the extract, the organic carbon content of the humic acid fraction (HA + FA) was determined. The remainder was acidified to pH 1.0–1.5 and left overnight to precipitate the humic acids (HA). After the separation of HA, the organic carbon content of the fulvic acid (FA) fraction was determined. The sediment remaining from the extraction was mixed with 50 $cm^3$ of $H_2SO_4$ (0.5 mol·$dm^{-3}$) and heated in a water bath for 1 h at 80 °C. The obtained extract was discarded. The remaining sediment was dried at a temperature of 80 °C to a constant weight (24 h), and the organic carbon content of the humin fraction (HU) was determined. The concentration of organic carbon in the extract solutions after neutralization was determined using a TOC-VCPN analyzer (Shimadzu). The organic carbon content in the extraction residue was determined with a CNS elemental analyzer, Flash EA 1112 (Finnigan Mat). The organic carbon content of the humic acid fraction (HA) was calculated from the difference between the organic carbon content of the humic acid fraction (FA + HA) and the

organic carbon content of the fulvic acid fraction (FA). The organic carbon content of humic substances (HS) was the sum of FA, HA, and HU. The sediment samples were analyzed in three replicates for which the relative standard deviations (%RSDs) were less than 10% for all HS fractions. An internal laboratory reference material (LRM) was used for quality control, in which the total organic carbon content of HS was determined by an accredited laboratory as part of interlaboratory comparative tests. The recovery of the HS fraction [mgC·g$^{-1}$] achieved by the sequential extraction technique was assessed by comparing the sum of HS fractions extracted in two stages (FA + HA and HU) with the total content of HS determined in an internal standard. The percentage of the HS recovery ranged from 91.4% to 107.5% (for 5 control samples).

### 2.3. Ecological Risk Assessment

The potential threat to aquatic organisms posed by heavy metals accumulated due to the amount of their accumulation in bottom sediments was determined based on the Sediment Quality Guidelines (SQGs) using numerical indicators of sediment quality TEC (threshold effect concentration) and PEC (probable effect concentration) [17,19,21]. The HQ risk factor was calculated based on the content of the individual metals and the corresponding PEC value [38].

$$HQ = \frac{C_{Me}}{PEC} \tag{1}$$

$C_{Me}$ is the determined content of the given metal in the sediment [mg·kg$^{-1}$] and PEC is the probable effect concentration. When HQ is <1, there will be no frequent adverse ecological effects due to the contamination of sediment with a given heavy metal [38].

The calculation of the mean risk factor (PEC$_q$) based on the total content of heavy metals determined in the sediment allows the combined toxic effect of many pollutants on aquatic organisms to be assessed.

$$PEC_q = \frac{\sum \frac{C_{Me}}{PEC}}{n} \tag{2}$$

$C_{Me}$ is the determined content of a given metal in the sediment [mg·kg$^{-1}$], PEC is the probable effect concentration, and n is the number of heavy metals determined. The criteria developed by Ingersoll et al. [39] and Perrodin et al. [40] were used to assess the ecological risk.

### 2.4. Statistical Analysis

The mean values of the determined parameters and standard deviations were calculated with the use of MS Excel 2013. The relationships between the analyzed parameters (heavy metals and OM fractions) were assessed by simple regression analysis with a given significance level of 0.05. The calculations were made with the use of Microsoft Excel 2019 for Windows 10 (Microsoft Corporation, Redmond, WA, USA) and Statistica 13.0 PL (StatSoft Poland) programs.

## 3. Results and Discussion

### 3.1. Granulometric Composition of Bottom Sediments

The bottom sediments of small retention reservoirs, depending on the geological structure of the catchment, differed in grain size distribution (Table 2).

The sediments of the Cierpisz, Brzóza Królewska, and Kamionka reservoirs had a large share of the sand fraction (77.8–93.7%). The share of the silt fraction in the sediments of these reservoirs ranged from 6.26 to 19.8%. The sediments of these reservoirs were characterized by practically no clay fraction, except for the inflow region of the Brzóza Królewska reservoir. The varied geological structure of the Tarlaka stream catchment area and the river dregs introduced with the waters of the main tributary were responsible for a slight content of clay fraction (2.46%), a higher silt fraction (19.8%), and a lower sand amount in this area of the Brzóza Królewska reservoir. The sediments of the Nowa Wieś

and Blizne reservoirs clearly differed from the above-mentioned three reservoirs in terms of grain size composition. In the sediments of the Nowa Wieś reservoir, the silt fraction slightly predominated (49.9; 50.2%, respectively, for the site near the inflow and the dam) over the sand fraction (40.8; 38.8%). There was also a clay fraction (9.26; 11.0%). In the sediments of the Blizne reservoir, the silt fraction was the most dominant (63.1; 70.5%). Compared to the sediments of other reservoirs, the highest share was also the clay fraction (22.1; 15.9%), and the smallest was the sand fraction (12.7; 13.5%). The water retention time (even a dozen or so times longer) in this reservoir favors the sedimentation of fine particles (the size of silt and clay), which has an impact on the different physicochemical composition of the sediments.

**Table 2.** The granulometric composition of bottom sediments of the analyzed small retention reservoirs.

| Site | Granulometric Fractions [%] | | |
| :---: | :---: | :---: | :---: |
| | Clay < 0.002 mm | Silt 0.002–0.063 mm | Sand 0.063–2.0 mm |
| CI1 | 0.01 | 10.9 | 89.1 |
| CI2 | 0.01 | 9.79 | 90.2 |
| BK1 | 2.46 | 19.8 | 77.8 |
| BK2 | 0.01 | 7.06 | 92.9 |
| NW1 | 9.26 | 49.9 | 40.8 |
| NW2 | 11.0 | 50.2 | 38.8 |
| KA1 | 0.01 | 9.20 | 90.8 |
| KA2 | 0.01 | 6.26 | 93.7 |
| KA3 | 0.01 | 8.15 | 91.8 |
| BL1 | 22.1 | 63.1 | 12.7 |
| BL2 | 15.9 | 70.5 | 13.5 |

*3.2. Sediment Contamination Level and Ecological Risk*

The sediments of the Cierpisz reservoir in the area of the dam (CI2) were characterized by the lowest content of cadmium and chromium. The lowest contents of copper, nickel, and zinc were found in the sediments of the Brzóza Królewska reservoir also in the vicinity of the dam (BK2). In the case of lead, the lowest content was recorded in the sediments of the central part of the Kamionka reservoir (KA2). The sediments of the Nowa Wieś reservoir turned out to be the most contaminated with heavy metals. The highest contents of Cd, Pb, Cr, Ni, and Zn were found in the area of the dam (NW2), while Cu was found near the inflow of this reservoir (NW1). It was observed that the sediments of the Cierpisz, Brzóza Królewska, and Kamionka reservoirs, characterized by a high proportion of sand fraction (>78%) and a slightly acidic pH (<6.5), were generally characterized by a much lower accumulation of heavy metals (Tables 2 and 3).

In the sediments of the Cierpisz and Brzóza Królewska reservoirs, a decrease in the accumulation of heavy metals was observed along with the water flow through the reservoir from the inflow to the dam, which proves the greater impact of the total catchment as these pollutants are introduced mainly along with the tributary waters. In the Nowa Wieś and Blizne reservoirs, the catchment area directly surrounding the objects had a greater impact on sediment pollution; hence, their higher accumulation was usually in the vicinity of the dam. In the Kamionka reservoir, the differences in the amount of metal accumulation indicated a clear influence of the total catchment near the tributary and the immediate catchment near the dam, as the lowest concentrations of all metals (except Cu) were observed in the central part of the reservoir.

The risk analysis of the harmful influence of heavy metals in sediments on aquatic organisms based on the ecotoxicological criterion [17,19] showed that the sediments of the Nowa Wieś reservoir constituted the greatest potential threat to the biosphere. In the sediments of this reservoir all the heavy metals tested exceeded the TEC threshold values. In the sediments of the Blizne reservoir in the area of the dam (BL2), the contents

of cadmium, chromium, nickel, and lead exceeded the TEC threshold values. With the exception of the Cierpisz reservoir in the dam area (CI2), the Cd content in the sediments of all reservoirs exceeded the TEC threshold values. None of the analyzed heavy metals in the sediments of the studied reservoirs exceeded the PEC values. This means that the harmful effect of a given metal on aquatic organisms, especially benthic organisms, may occur (>TEC), but it will not occur continuously (<PEC).

**Table 3.** Heavy metal and organic matter content in bottom sediments of the analyzed small retention reservoirs (average ± SD) and pH range of the sediments. TEC and PEC values for given metals [17].

| Site n = 9 | Cd | Pb | Cr | Cu [mg·kg⁻¹ dw] | Ni | Zn | OM [%] | $C_{org.}$ [mg·g⁻¹ dw] | $pH_{KCl}$ [–] |
|---|---|---|---|---|---|---|---|---|---|
| CI1 | 1.50 ± 0.27 | 9.44 ± 4.60 | 4.93 ± 1.84 | 2.77 ± 0.92 | 2.62 ± 0.76 | 27.2 ± 6.1 | 1.78 ± 1.04 | 6.31 ± 4.28 | 3.84–5.29 |
| CI2 | 0.88 ± 0.45 | 6.02 ± 2.31 | 2.75 ± 1.07 | 1.82 ± 1.58 | 2.03 ± 0.92 | 17.6 ± 6.9 | 1.38 ± 0.47 | 5.18 ± 2.69 | 4.23–5.55 |
| BK1 | 2.70 ± 0.70 | 35.1 ± 15.6 | 12.6 ± 6.8 | 14.4 ± 5.5 | 7.75 ± 4.24 | 42.1 ± 20.9 | 6.48 ± 3.73 | 33.5 ± 19.9 | 4.72–5.38 |
| BK2 | 1.16 ± 0.15 | 5.80 ± 3.51 | 3.50 ± 1.28 | 0.836 ± 0.523 | 1.35 ± 0.53 | 9.91 ± 4.32 | 1.00 ± 0.51 | 2.96 ± 1.84 | 4.32–6.41 |
| NW1 | 3.18 ± 0.96 | 56.3 ± 16.3 | 47.1 ± 14.9 | 47.5 ± 16.6 | 34.8 ± 11.0 | 227 ± 95 | 12.4 ± 4.2 | 43.5 ± 15.7 | 5.30–7.34 |
| NW2 | 3.49 ± 0.70 | 69.7 ± 9.3 | 59.8 ± 7.5 | 43.6 ± 5.1 | 45.4 ± 5.5 | 237 ± 54 | 14.0 ± 2.7 | 42.1 ± 9.1 | 5.86–7.58 |
| KA1 | 1.85 ± 0.78 | 16.4 ± 10.3 | 13.1 ± 7.6 | 3.58 ± 2.01 | 4.40 ± 2.37 | 31.5 ± 15.1 | 2.75 ± 2.47 | 14.1 ± 12.9 | 4.24–5.44 |
| KA2 | 1.15 ± 0.45 | 5.01 ± 2.67 | 4.86 ± 3.36 | 3.47 ± 3.22 | 2.24 ± 1.35 | 17.5 ± 7.6 | 2.19 ± 2.02 | 10.6 ± 10.8 | 4.78–5.43 |
| KA3 | 1.59 ± 0.30 | 10.1 ± 5.0 | 17.7 ± 9.5 | 1.53 ± 0.46 | 3.21 ± 1.15 | 29.3 ± 11.6 | 2.47 ± 1.02 | 10.6 ± 7.4 | 4.45–5.99 |
| BL1 | 1.64 ± 0.60 | 34.8 ± 7.4 | 40.6 ± 2.8 | 18.5 ± 4.2 | 22.4 ± 2.7 | 50.1 ± 4.0 | 7.16 ± 1.39 | 30.6 ± 9.7 | 7.78–8.19 |
| BL2 | 2.55 ± 0.54 | 49.1 ± 9.3 | 48.4 ± 4.2 | 28.0 ± 5.9 | 31.5 ± 3.0 | 53.2 ± 5.7 | 5.83 ± 2.34 | 20.9 ± 18.6 | 6.54–8.14 |
| TEC | 0.99 | 35.8 | 43.4 | 31.6 | 22.7 | 121 | | | |
| PEC | 4.98 | 128 | 111 | 149 | 48.6 | 459 | | | |

n—number of samples collected from each site.

The exceeding of the TEC threshold value by cadmium concentration in sediments was also observed in another small retention reservoir in Podkarpacie [41]. Cadmium is highly toxic to living organisms, even in low concentrations [9,14,26,42]; therefore, low limit values for the content of this element were used in the assessment criteria. Its toxic effect consists of disrupting the work of many organs, causing respiratory diseases, and being a potential carcinogen [42,43]. In certain concentrations in the aquatic environment, it disrupts the processes of photosynthesis and the mineralization of organic matter, similarly to other heavy metals. Soils contaminated with this metal are considered to be a significant source of cadmium in agricultural catchments. The long-term and widespread use of artificial fertilizers (e.g., superphosphates) and plant protection products has led to soil contamination mainly with cadmium but also with copper and chromium, from where they are discharged into rivers and water reservoirs along with surface runoff [44–46]. Increased cadmium content was found in sedimentary rocks. The concentration of this element was an average of 0.22 mg·kg⁻¹ in diatomites, 0.05 mg·kg⁻¹ in gaizes, 0.4 mg·kg⁻¹ in opoka-rocks, and 2.23 mg·kg⁻¹ in light opoka-rocks. In the process of weathering, this element is easily activated, and it then binds with clay minerals, iron hydroxides and organic matter [47].

The ecological risk analysis carried out on the basis of the average $PEC_q$ risk factor for six metals (Table 4) showed that the sediments of the Cierpisz and Brzóza Królewska reservoirs in the vicinity of the dam (BK2) and Kamionka in the central part (KA2) can be considered as not showing a potential threat to hydrobionts ($PEC_q < 0.1$, i.e., harmless). The sediments of the Blizne and Brzóza Królewska reservoirs near the tributary (BK1) and the Kamionka near the tributary and the dam (KA1 and KA3) may already pose a potential threat to benthic fauna to a slight degree ($0.1 < PEC_q < 0.5$). According to the criterion of Ingersoll et al. [39], sediments of the Nowa Wieś reservoir ($PEC_q = 0.51$ and 0.59 for NW1 and NW2, respectively) pose a threat to aquatic organisms due to the toxic effects of heavy metals contained in them ($0.5 < PEC_q < 1.0$). According to the criterion of Perrodin et al. [40], the average risk factor $PEC_q > 0.5$ indicates a high potential risk of toxicity of the Nowa Wieś reservoir sediments for aquatic organisms feeding at the bottom.

Bottom sediments of other reservoirs ($PEC_q < 0.5$) showed a low level of potential toxicity (or non-toxicity) to benthic organisms. The analysis of the ecological risk of the Rzeszów reservoir sediments (another small retention reservoir in this region of Poland) showed that, in the region of the tributary and the dam, they were highly toxic to benthic fauna ($PEC_q > 0.5$) due to contamination with heavy metals and PAHs [48].

**Table 4.** Hazard quotients (HQ) calculated for the individual metals and mean risk quotients (PECq).

| Site | Cd | Pb | Cr | Cu | Ni | Zn | PECq |
|------|------|------|------|------|------|------|------|
| CI1 | 0.30 | 0.07 | 0.04 | 0.02 | 0.05 | 0.06 | 0.09 |
| CI2 | 0.18 | 0.05 | 0.02 | 0.01 | 0.04 | 0.04 | 0.06 |
| BK1 | 0.54 | 0.27 | 0.11 | 0.10 | 0.16 | 0.09 | 0.21 |
| BK2 | 0.23 | 0.05 | 0.03 | 0.01 | 0.03 | 0.02 | 0.06 |
| NW1 | 0.64 | 0.44 | 0.42 | 0.32 | 0.72 | 0.49 | 0.51 |
| NW2 | 0.70 | 0.54 | 0.54 | 0.29 | 0.93 | 0.52 | 0.59 |
| KA1 | 0.37 | 0.13 | 0.12 | 0.02 | 0.09 | 0.07 | 0.13 |
| KA2 | 0.23 | 0.04 | 0.04 | 0.02 | 0.05 | 0.04 | 0.07 |
| KA3 | 0.32 | 0.08 | 0.16 | 0.01 | 0.07 | 0.06 | 0.12 |
| BL1 | 0.33 | 0.27 | 0.37 | 0.12 | 0.46 | 0.11 | 0.28 |
| BL2 | 0.51 | 0.38 | 0.44 | 0.19 | 0.65 | 0.12 | 0.38 |

### 3.3. Organic Matter and Humic Substances Distribution

An important role in the process of the permanent accumulation of pollutants in bottom sediments may be played not only by the general enrichment but also by the nature of organic matter and its susceptibility to decomposition [15]. The sediments of the Nowa Wieś reservoir near the dam (NW2) were the most enriched in organic matter (14%) and HS (39.4 mgC·g$^{-1}$ dw). The least enriched in OM and HS (1.0% and 2.78 mgC·g$^{-1}$ dw, respectively) were the sediments of the Brzóza Królewska reservoir, also in the vicinity of the dam (BK2) (Table 3 and Figure 2). A similar situation occurred in the case of FA (7.25 mgC·g$^{-1}$ dw and 0.69 mgC·g$^{-1}$ dw, respectively).

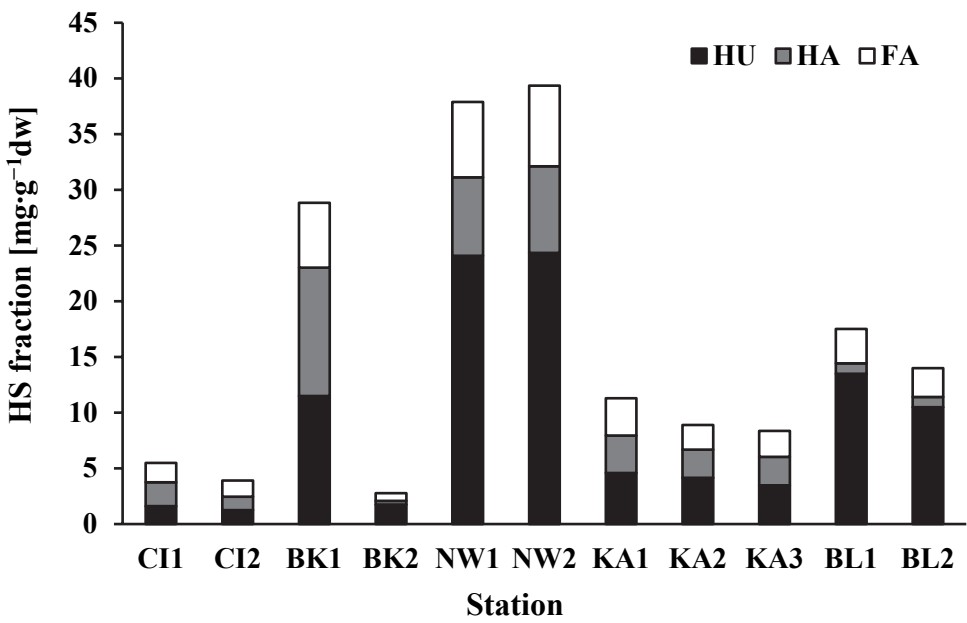

**Figure 2.** The content of fractions of humic substances (HS = HU + HA + FA), fulvic acids (FA), humic acids (HA), and humins (HU) in the bottom sediments of the analyzed reservoirs.

Both the highest and the lowest content of HA was observed in the sediments of the Brzóza Królewska reservoir (11.5 mgC·g$^{-1}$ dw—BK1 and 0.33 mgC·g$^{-1}$ dw—BK2).

The highest content of humins (HU) (24.3 mgC·g$^{-1}$ dw) was also found in the sediments of the Nowa Wieś reservoir, near the dam (NW2). The lowest content of this fraction (1.24 mgC·g$^{-1}$ dw) was found in the sediments of the Cierpisz reservoir, near the dam (CI2). The carbon of humic substances constituted from 59 to 95% of the organic carbon in the sediments of the studied reservoirs (Figure 3). Usually, the highest percentage of organic carbon was that of the HU fraction (except for Cl1, Cl2, and KA1). A clearly dominant share of the HU fraction (46–58%) was present in the sediments of two reservoirs, Nowa Wieś and Blizne.

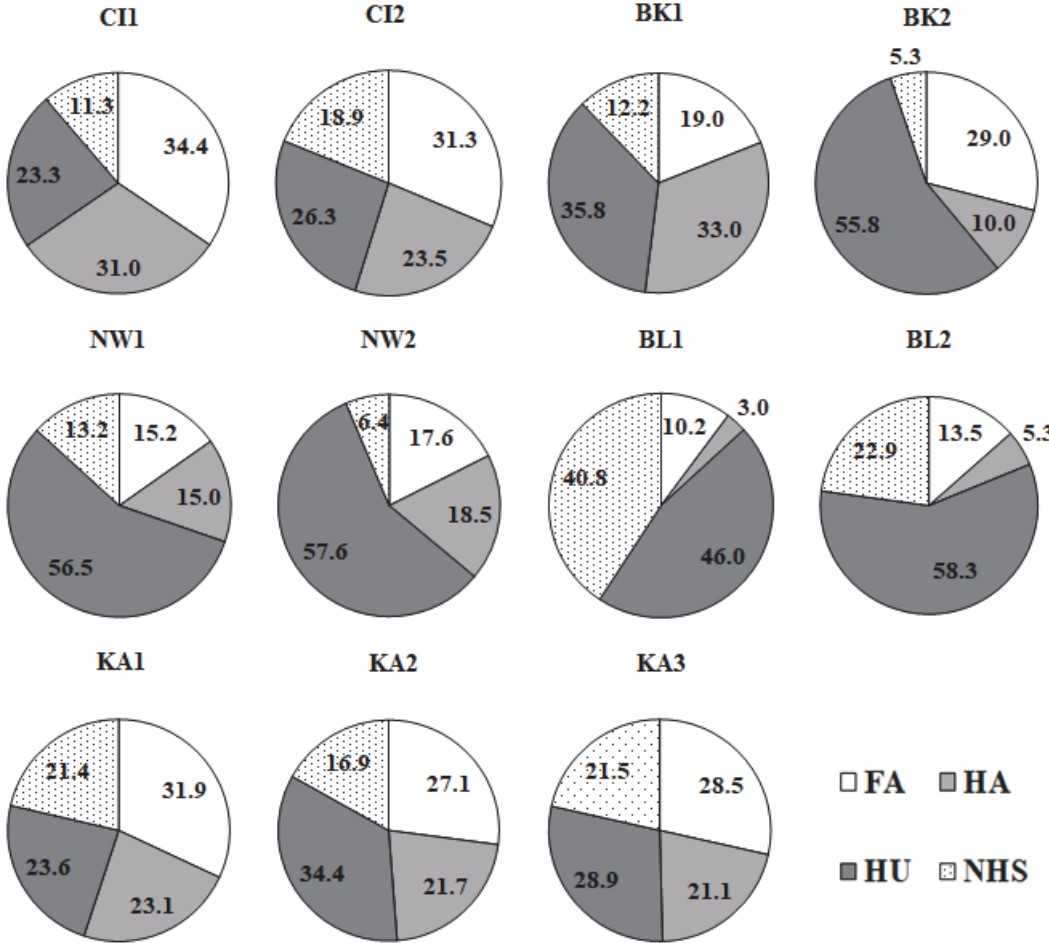

**Figure 3.** The share of individual HS fractions [%] in C$_{org.}$ in the bottom sediments of the studied small retention reservoirs. NHS, non-humic substances.

Sediments with a high proportion of sand fraction were usually less enriched in HS and its fractions, except for slightly more silt sediments (19.8% silt fraction) from the Brzóza Królewska reservoir near the tributary (BK2). In general, the sediments near the tributary were characterized by a higher content of HS and their fractions, as well as OM (except for the sediments of the Nowa Wieś reservoir). This proves a significant share of external sources in the accumulation of this type of matter in the sediments of small retention reservoirs. The sediments of the Nowa Wieś reservoir were much more enriched in organic matter, HS, FA, and HU fractions compared to the sediments of the remaining reservoirs.

A significant load of organic matter, mainly humic substances, is introduced into surface waters by surface runoff from wetlands and peat bogs [24,25,49], which are abundant in the catchment area of the Nowa Wieś reservoir. Fulvic and humic acids may occur in various amounts depending on the type of soil in the catchment area. It is believed that fulvic acids dominate in soils in forest areas, while humic acids dominate in meadow and peat soils [22]. A large share of forest soil occurs in the catchments of the Cierpisz,

Kamionka, and Brzóza Królewska reservoirs, which is generally reflected in the advantage of the FA fraction over HA (except for BK1). In the sediments of the Nowa Wieś reservoir, the average share of FA and HA fractions was evenly distributed. On the other hand, a significant share of meadow soils in the Blizne reservoir catchment did not contribute to the dominance of the HA fraction compared to FA in the sediments of this object.

### 3.4. Effect of Organic Matter on the Accumulation of Heavy Metals

The comparison of the studied reservoirs in terms of the increasing average OM content and the risk index of their ecotoxic effect on aquatic organisms ($PEC_q$) (Figure 4) indicates a possible relationship between the accumulation of heavy metals and organic matter in the bottom sediments of small retention reservoirs. According to El-Radaideh et al. [50], the content of organic matter (total) and the grain size of the sediment are the main factors determining the concentration of heavy metals. Many authors emphasize the relationship of heavy metals with the enrichment in OM in the sediments of dam reservoirs [11,12,51]. The smallest granulometric fractions show a high ability to adsorb metal cations [1,17,52].

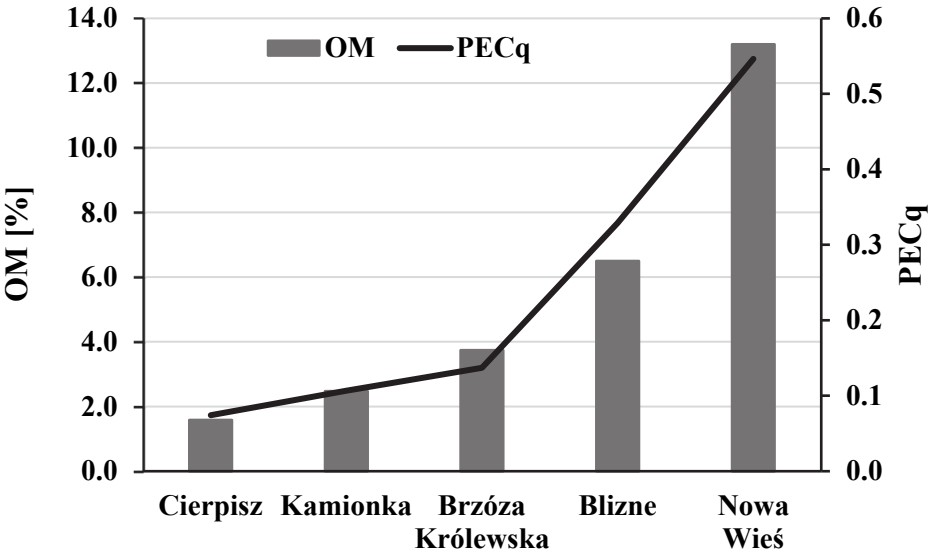

**Figure 4.** The relationship between the risk index of the ecotoxic effect on aquatic organisms ($PEC_q$) and the content of organic matter in the sediments of small retention reservoirs.

It was observed that the sediments with a higher proportion of sand fraction (>50%) were characterized by a lower level of contamination with heavy metals and lower enrichment in OM (1.58–3.74%). According to Hu et al. [53], the adsorption capacity of metals by solid particle fractions decreases with increasing grain size. Due to the content of clay minerals and organic matter, fine-grained clay and silt fractions have the ability to accumulate more pollutants than coarse particles of sandy sediments [26]. The content of the sandy fraction in the bottom sediments of the studied reservoirs did not seem to have a direct impact on the accumulation of metals, but it was clearly related to the enrichment in OM. According to Gierszewski [54], organic matter is the most important factor influencing the accumulation of heavy metals in the sediments of dam reservoirs. In the Włocławek dam reservoir (Poland), the highest accumulation of heavy metals and OM was recorded in the sediments near the dam [54]. In large dam reservoirs, the greatest accumulation of organic matter was observed in the sediments of the lake zone, due to the more favorable conditions for internal production and sedimentation [6,12,50]. On the other hand, the lake sediments most enriched in OM occur in the zone with the greatest depth [55]. In all the objects, except for the Nowa Wieś reservoir, higher concentrations of organic matter and HS fractions occurred in the sediments near the tributary. In the case of heavy metals, such a situation occurred in the sandier sediments of the Cierpisz, Brzóza Królewska, and

Kamionka reservoirs. This indicates a significant influence of the watercourse catchment on which the reservoir was created.

Taking into account the sediments of all the considered reservoirs, statistically significant relationships ($p < 0.05$) were found between the content of all analyzed heavy metals and the content of total organic matter and all its humic fractions. However, considering each reservoir individually, it was observed that, only in the sandy sediments of the Brzóza Królewska reservoir, both OM and all HS fractions had a statistically significant effect on the accumulation of heavy metals (Table 5). In the sediments of the Nowa Wieś reservoir, a statistically significant effect on the accumulation of metals was found in the case of OM, FA, HA, SH (without Cd), and HU (without Cd, Pb, Cr, Ni) (Table 6).

**Table 5.** Relationships between OM and HS fractions and heavy metals in the bottom sediments of reservoirs with sand fractions >50%.

| r | Cierpisz (n = 18) | | | | | Brzóza Królewska (n = 18) | | | | | Kamionka (n = 27) | | | | |
|---|---|---|---|---|---|---|---|---|---|---|---|---|---|---|---|
| | OM | HS | FA | HA | HU | OM | HS | FA | HA | HU | OM | HS | FA | HA | HU |
| Cd | - | - | - | - | - | 0.75 | 0.74 | 0.78 | 0.73 | 0.70 | 0.41 | 0.48 | 0.55 | 0.45 | 0.45 |
| Pb | - | - | - | - | - | 0.90 | 0.93 | 0.90 | 0.93 | 0.89 | - | - | - | - | - |
| Cr | 0.71 | 0.76 | 0.53 | 0.80 | 0.67 | 0.96 | 0.96 | 0.95 | 0.95 | 0.93 | 0.47 | - | - | - | - |
| Cu | - | - | - | - | - | 0.78 | 0.79 | 0.82 | 0.70 | 0.84 | - | - | - | - | - |
| Ni | 0.58 | 0.59 | - | 0.55 | 0.60 | 0.90 | 0.91 | 0.89 | 0.91 | 0.87 | 0.63 | 0.54 | 0.50 | 0.54 | 0.54 |
| Zn | 0.58 | 0.60 | 0.52 | 0.61 | 0.51 | 0.97 | 0.97 | 0.95 | 0.97 | 0.94 | 0.73 | 0.70 | 0.69 | 0.71 | 0.66 |

r—Pearson correlation coefficient; double underline $p < 0.001$; single underline $p < 0.01$; no underline $p < 0.05$; - non-significant correlation.

**Table 6.** Relationships between OM and HS fractions and heavy metals in the bottom sediments of reservoirs with silt fractions >50%.

| r | Nowa Wieś (n = 18) | | | | | Blizne (n = 18) | | | | |
|---|---|---|---|---|---|---|---|---|---|---|
| | OM | HS | FA | HA | HU | OM | HS | FA | HA | HU |
| Cd | 0.62 | - | 0.58 | 0.64 | - | - | - | - | - | - |
| Pb | 0.81 | 0.60 | 0.65 | 0.64 | - | - | - | - | - | - |
| Cr | 0.78 | 0.56 | 0.70 | 0.47 | - | - | - | - | - | - |
| Cu | 0.71 | 0.77 | 0.85 | 0.66 | 0.55 | - | - | - | - | - |
| Ni | 0.80 | 0.54 | 0.62 | 0.50 | - | - | - | - | - | - |
| Zn | 0.86 | 0.85 | 0.84 | 0.84 | 0.57 | - | - | - | - | - |

r—Pearson correlation coefficient; double underline $p < 0.001$; single underline $p < 0.01$; no underline $p < 0.05$; - non-significant correlation.

In the Kamionka reservoir, the organic matter was statistically significantly associated with the contents of Cd, Cr, Ni, and Zn in the sediments. Humic substances (in total) and their fractions were statistically significantly correlated with the contents of Cd, Ni, and Zn. In the sediments of the Cierpisz reservoir, the contents of Cr, Ni, and Zn were statistically significantly associated with the contents of OM, HS, HA, and HU. The FA fraction significantly correlated only with Cr and Zn. In the case of the sediments of the Blizne reservoir, no statistically significant correlations between these parameters were observed. Correlations between the content of the (total) organic matter and selected metals (e.g., Cu, Zn, and Pb) were already observed in the bottom sediments of other reservoirs [17,21,56]. De la Rosa et al. [29] showed that HA has the ability to bind significant amounts of Cu and Zn, and they also indicated the importance of HS for the transport and binding of trace elements in bottom sediments. The retention capacity of sedimentary HA that they demonstrated was as follows: Cu >>> Zn >> Cr $\geq$ Ni $\geq$ Pb. The correlation analysis showed that statistically significant relationships between Cu and HA occurred only in two reservoirs, Brzóza Królewska and Nowa Wieś. On the other hand, Zn and Ni significantly correlated with HA in all reservoirs, except for the Blizne reservoir.

The analysis of the above-mentioned correlations did not clearly confirm that the organic matter was a parameter that had a decisive influence on the cumulation of heavy metals in the sediments of small retention reservoirs. However, such a relationship occurred for some or all of the analyzed metals and HS fractions, except for one reservoir. In reservoirs where the influence of organic matter or its fraction were not statistically confirmed for all (Blizne) or some (Cierpisz and Kamionka) heavy metals, there is probably an overlap of the influence of various factors, i.e., the composition of bottom sediments (OM, HS), the type of grain size distribution, and the nature and development of the catchment area. The anthropogenic influence of the catchment area seems to be the most important, but one should distinguish between the influence of the main tributary catchment, the direct catchment area, and the total catchment area. Heavy metals are pollutants of anthropogenic origin, while humic substances are of natural origin. Humic substances can form strong complexes with metal ions, influencing their speciation, mobilization, and toxicity in the environment [23]. The correlations between these parameters may indicate the formation of humus-metal connections in the sediments, but they may also indicate their introduction in the form of salt and complex connections along with surface runoff into the reservoir (common source). The lack of correlation precludes both of these options. Pasture and agricultural management of the direct catchment area of the Blizne reservoir and production inside the reservoir determine the enrichment of sediments in OM and HS, while the source of heavy metals is probably mainly dust from fuel combustion, which is a carrier of pollutions of long-range sources. Hence, there is no link between these parameters.

## 4. Conclusions

Investigations of bottom sediments of small retention reservoirs with a diversified nature of the catchment area and anthropopressure did not clearly confirm the relationships between the content of humic substances and their fractions and the accumulation of heavy metals. However, it was shown that, in these types of objects, there is a connection between the enrichment of sediments with organic matter and the increased risk of their ecotoxic impact on aquatic organisms, and this is determined by excessive exposure to heavy metal contamination.

The content of the sandy fraction had no obvious effect on the accumulation of metals in bottom sediments, but in the studied reservoirs, it was to some extent related to the enrichment in organic matter. Hence, sediments with a high proportion of sand were characterized by a low content of organic matter and heavy metals.

The sediments of the reservoir most exposed to anthropopressure from the catchment (Nowa Wieś) were characterized by a high potential risk of toxicity to aquatic organisms feeding at the bottom. The bottom sediments of the remaining reservoirs showed a low level of potential toxicity (or non-toxicity) to benthic organisms. The catchment determines the number of heavy metals introduced into the reservoirs, but their further fate depends on the possibility of their retention in the sediments, including the ability to chemically bind them. Humic-metal connections may arise not only in the reservoir but also within the catchment area and migrate to the reservoir in this form.

Summing up, the accumulation of heavy metals in the sediments of small retention reservoirs depends both on the concentration and nature of organic matter (including the content of humic substances and their fractions), as well as on the nature and development of the catchment area. The influence of the above-mentioned factors is a complex phenomenon, and their interaction determines the differentiation of the sediments in terms of contamination with heavy metals. However, the most important factor remains the exposure to pollution, i.e., the supply of anthropogenic substances from the immediate catchment area and through incoming watercourses.

**Author Contributions:** Conceptualization, L.B.; methodology, L.B., R.G.-R., A.P. and J.C.; formal analysis, L.B., R.G.-R. and A.P.; investigation, L.B., R.G.-R., A.P. and J.C.; resources, R.G.-R. and J.C.; data curation, L.B.; writing—original draft preparation, L.B.; writing—review and editing, L.B. and A.P.; supervision, R.G.-R. and J.C.; project administration, L.B.; funding acquisition, L.B. (co-author

of the grant application and main contractor). All authors have read and agreed to the published version of the manuscript.

**Funding:** Part of the research was funded by the National Science Center (Poland), grant number 2011/03/B/ST10/04998.

**Data Availability Statement:** The authors confirm that the data supporting the findings of this study are available within the article. Raw data that support the findings of this study are available from the corresponding authors, upon reasonable request.

**Acknowledgments:** We would like to thank our colleagues from the department laboratory for their support and help in sampling and laboratory analysis.

**Conflicts of Interest:** The authors declare no conflict of interest.

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
