# Peer review of "Heavy Metal Accumulation in Sediments of Small Retention Reservoirs—Ecological Risk and the Impact of Humic Substances Distribution"

_resources, doi:10.3390/resources11120113_

Round 1

Reviewer 1 Report

This study has provided important information on the assessment of metal toxicity in the sediments of the reservoirs and the effects on the organisms and the environment. The manuscript is well structured with appropriate tables and figures. However, the expressions used in some sentences are ambiguous and need to be improved for clarity.

There are some questions need to be addressed as follows:

1)      It is commended that the authors have used an appropriate CRM for quality control and method validation. It would be useful if the data of recoveries of these metals are included in the discussion for precision and accuracy. In lines 131-132, the sentence starts with “The certified reference material CRM…..control the quality…..analyzes”, could be re-written as “The certified……….CRM 16-050 was used for quality control and method validation…. and the recoveries of these trace elements satisfactory (please provide data).

2)      Lines 136-154, no reference materials were mentioned for quality control and method validation. It would be useful to discuss what standards and calibrations were used for method validation for analysis of humic substances.

3)      In line 167, equation 2, in line 169, is the “n” value indicating the amount or the number?

4)      In Table 3, “n=9” under site, the n value needs to be clearly stated that indicates the number of samples collected from each site. At the bottom of the, what does the forward “/” in “/PEC indicate?

5)      In Tables 5 & 6, the statistical significance of Pearson correlation needs to be identified with each of the p values (<0.001, <0.01, <0.05).

6)      The conclusions (line 403) require re-write for clarity and to be more convincing on the findings that reflect the objectives of the study.

It would be helpful if the manuscript is to be proof read by native or near-native English speaker for improvement.

Reviewer 2 Report

Dear author,

the MS entitled "Heavy metals accumulation in sediments of small retention reservoirs - Ecological risk and the impact of humic substances distribution" is interesting to the reader and could be published in the journal "Resources" after minor revision as follows:

line 108: sampling is conducted 2013-14 and direct afterwords the analysis of the samples? please specify in the MS.

line 128: concentrated HNO3 digestion does not give the total metal content. The efficiency of such digestion is only about 80% of the total metal concentration of some elements like Cd, Ni, and Cr. Authors should point this in the MS

line 132: Authors should provite if possible the data on QA/QC

lines 142 &151: humic not humin
